# Mutation in the CX3C Motif of G Protein Disrupts Its Interaction with Heparan Sulfate: A Calorimetric, Spectroscopic, and Molecular Docking Study

**DOI:** 10.3390/ijms23041950

**Published:** 2022-02-09

**Authors:** Abu Hamza, Abdus Samad, Zahoor Ahmad Parray, Sajda Ara, Anwar Ahmed, Fahad N. Almajhdi, Tajamul Hussain, Asimul Islam, Shama Parveen

**Affiliations:** 1Centre for Interdisciplinary Research in Basic Sciences, Jamia Millia Islamia, New Delhi 110025, India; abuhamza830@gmail.com (A.H.); a.samad0195@gmail.com (A.S.); zaparray@gmail.com (Z.A.P.); sajdaarabiotech.131@gmail.com (S.A.); 2Center of Excellence in Biotechnology Research, College of Science, King Saud University, Riyadh 11451, Saudi Arabia; anahmed@ksu.edu.sa (A.A.); majhdi@ksu.edu.sa (F.N.A.); thussain@ksu.edu.sa (T.H.); 3Department of Botany & Microbiology, College of Science, King Saud University, Riyadh 11451, Saudi Arabia

**Keywords:** RSV, G protein, CX3C motif, glycosaminoglycan, heparan sulfate, fluorescence binding, isothermal titration calorimetry, molecular docking

## Abstract

Respiratory syncytial virus (RSV) is the leading cause of lower respiratory tract infection in children and infants. To date, there is no effective vaccine available against RSV. Heparan sulfate is a type of glycosaminoglycan that aids in the attachment of the RSV to the host cell membrane via the G protein. In the present study, the effect of amino acid substitution on the structure and stability of the ectodomain G protein was studied. Further, it was investigated whether mutation (K117A) in the CX3C motif of G protein alters the binding with heparan sulfate. The point mutation significantly affects the conformational stability of the G protein. The mutant protein showed a low binding affinity with heparan sulfate as compared to the wild-type G protein, as determined by fluorescence quenching, isothermal titration calorimetry (ITC), and molecular docking studies. The low binding affinity and decreased stability suggested that this mutation may play an important role in prevention of attachment of virion to the host cell receptors. Collectively, this investigation suggests that mutation in the CX3C motif of G protein may likely improve the efficacy and safety of the RSV vaccine.

## 1. Introduction

Respiratory syncytial virus (RSV), which belongs to the family *Pneumoviridae* and genus *Orthopneumovirus*, is the leading cause of acute lower respiratory infection in the elderly and children, especially those less than 2 years of age [1,2,3]. RSV is the main cause for the hospitalization of approximately 20 per 1000 infants, which leads to the deaths of around 6.6 per 1000 infants of less than 12 months [4]. A report articulated that globally, around 48,000 to 74,000 deaths occur every year due to RSV, mostly in the pediatric population [5]. Currently, no approved vaccine is available to prevent RSV infection. The only defensive measure is the injection of palivizumab, a monoclonal antibody specific to the viral fusion (F) glycoprotein [6]. The two envelope glycoproteins of RSV, G and F, are responsible for viral attachment and fusion, respectively. The G protein of RSV is a type II transmembrane glycoprotein and acts as an important target of host immune cell response. The size of the G protein varies in circulating strains from 282–321 aa. The G protein comprises three main domains: the cytoplasmic domain of amino acids 1–37, the transmembrane domain of amino acids 38–66, and the ectodomain region of amino acids 67–312. The ectodomain part of the G protein has an unglycosylated central conserved region. The two hypervariable mucin-like domains also present flanking to the central conserved region [7,8,9]. The central region contains 13 amino acid central conserved domains (CCDs) of amino acids 164–176 that are highly conserved in all strains of RSV [10,11,12]. The four cysteine residues at the amino acid positions 173, 176, 182, and 186 form the cysteine noose, which is bound together via two disulfide bonds between Cys173–Cys186 and Cys176–Cys182 [13]. This region serves as a putative receptor-binding site and is slightly hydrophobic [11]. Two hypervariable regions are present flanking the cysteine regions that are similar to the amino acid composition of mucin, a protein secreted by the epithelial cell. It has a high content of amino acids such as serine, threonine, and proline, which serve as hotspots for mutations [14]. The mucins are host glycoproteins that act as a protective barrier on gastrointestinal, respiratory, and reproductive tracts. The third (Cys182) and fourth cysteine (Cys186) of the central conserved domain form the CX3C motif of amino acids 182–186. The CX3C motif of the G protein aids in the attachment of RSV to the vulnerable human airway epithelial (HAE) cells with the interaction of the chemokine receptor (CX3CR1) [13,15,16,17]. The G protein shows structural similarities with CX3C chemokine fractalkine and induces leukocyte chemotaxis in the in vitro condition [18]. Downstream from the cystine noose, the positively charged heparin-binding domain (HBD) is present in amino acids 184–198 of the G proteins. The HBD helps in the attachment of the RSV through cell surface glycosaminoglycans (GAGs) [19,20]. The GAGs (heparin and heparin sulfate) present on the surface of mammalian cells act as a receptor for RSV in different cell lines (such as Hep-2 cells). Previous studies showed that the basic amino acids of the HBD of the G protein helps in the attachment of RSV through negatively charged heparan sulfate present on the host cells [16,19]. Another study also reported that heparan sulfate, and to a lesser amount chondritin–4–sulfate, was also found to be involved in attachment of RSV to the membrane of host cell [21].

RSV infects many cell types, but mainly infects respiratory tract epithelial cells, which results in the expression of chemokine and cytokines, adaptive and innate immune responses, and pulmonary inflammation, which control viral replication and infection [22,23,24]. The monoclonal antibodies (mAbs), which targeted the G protein, neutralized the RSV infection in HAE cells and diminished the viral load in animal models [25,26]. The anti–G mAbs reduced mucus production, proinflammatory cytokines, and pulmonary inflammation, and reinstated the Th1/Th2 cytokine balance [27,28]. The mAbs targeting the CCD of the G protein were protective in prophylactic and post infection animal models [27,29].

The F and G glycoproteins both are targeted by the humoral immune response against RSV [30]. The human mAbs 3G12 and 3D3, which bind with the CCD, have been shown to strongly neutralize both subtypes of RSV. These antibodies bind to the G protein of RSV with high affinity, and protected mice from RSV infection [31,32]. Moreover, the CCD of the G protein has been shown to elicit a long-lasting and defensive antibody response in mice [33,34]. This type of study is directed to the isolation of a murine antibody (131–2G), which binds to the CCD and blocks the interaction of CX3CR1 with G protein, thus inhibiting viral attachment [18]. Previous studies showed that the T–cell epitope, which is found in amino acids 184–203 in the G protein, is involved in protective immunity against the RSV infection and induction of T-cell response and eosinophilia [35,36]. Thus, the CX3C motif and the CCD of the G protein are very important in designing an RSV vaccine. The immunization of mice with an RSV G protein or polypeptide having a CCD induces a protective antibody response [37,38], and likewise for the CX3C motif [39,40]. Recently, Bergeron and colleagues reported that a vaccine targeting the RSV G protein encoded with a central conserved domain induced antibodies, blocking the CX3C-CX3CR1 [41]. Since binding of the G protein to CX3CR1 through the CX3C motif is important to cause RSV infection; mutations in this motif might prevent disease. Previously, it was reported that mutating the CX3C motif in the G protein should improve the more effective and safer vaccine against RSV [37].

In the study presented here, we describe the structural and binding studies of a mutant G protein. We cloned and expressed the mutant ectodomain G protein in the bacterial system. Protein was purified from the bacterial cell pellet from the solubilized IBs using Ni–NTA affinity chromatography. We monitored changes in the tertiary structure in a wide range of pH, using fluorescence spectroscopy. Further, GdmCl- and urea-induced denaturation studies were performed to monitor the structural change and to measure stability of the protein. Isothermal titration calorimetry (ITC), fluorescence quenching, absorbance spectroscopy, and in silico approaches were exploited to determine the interaction of the mutant G protein with the heparan sulfate.

## 2. Results and Discussion

### 2.1. Strategy Used for the Generation of Mutation in the G Protein

Previous studies showed that the CX3C chemokine motif present in the central conserved region of G protein bound with the CX3CR1 receptor present on HAE cells [16]. The alterations of amino acid in this region are likely to affect the binding of the virion to the host cell receptors, therefore preventing RSV infection [37]. Based on this idea and reported studies, we mutated the positively charged lysine (K) amino acid residue to the neutrally charged alanine (A) in the CX3C motif. This substitutional mutation changed the CX3C motif (CK^183^SIC to CA^183^SIC) with respect to the full–length G protein (Figure 1A). Since we used the ectodomain region of the G protein for the present study, the mutated CX3C motif position was CK^117^SIC to CA^117^SIC (Figure 1B). We mutated the positively charged amino acid to the neutral amino acid to disrupt the interaction of the G protein with the host cell membrane. The previous study reported that the positively charged amino acids of the G protein interacted with negatively charged heparin/heparin sulfate molecules present on the host cell receptors. Therefore, we chose to mutate the lysine residues to alanine in the CX3C motif of the G protein.

### 2.2. Cloning, Expression, and Purification of the Mutant G Protein

The plasmid of the mutant G protein gene was transformed into the *E. coli* host cell strain of BL21 (DE3), and the isolated plasmid was analyzed using agarose gel electrophoresis (Figure 2A). A culture with induction of 0.5 mM IPTG was grown at 30 °C for 12 h for maximum protein expression. The inclusion bodies (IBs) were prepared from the harvested bacterial cell by sonication and centrifugation using lysis buffer and autoclaved Milli-Q water. Washing steps were used to eliminate the host protein, proteases, DNA, endotoxin, and nonspecific proteins [42]. The IBs were solubilized in CAPS buffer containing *N*–lauroylsarcosine. After solubilization and centrifugation, the soup was loaded on the Ni–NTA column, eluted with an increasing concentration of imidazole, and analyzed by SDS-PAGE. The desired protein fractions were eluted out with 100–200 mM of imidazole concentration. Further, the desired protein was dialyzed 4–5 times at 4 °C to refold the protein into the correct native conformation spontaneously. After dialysis, the protein solution was centrifuged and filtered using a 0.22 μm syringe filter to remove any precipitate formed during dialysis. The dialyzed protein was analyzed through SDS–PAGE, which indicated the purity of the protein (Figure 2B).

### 2.3. Structural and Conformational Stability Measurements of the Mutant G Protein

#### 2.3.1. Fluorescence Measurements

Fluorescence spectroscopy provides information about the tertiary structure of a protein due to the presence of aromatic amino acids, which are highly sensitive to the local environment. The change in the emission spectra of the mutant G protein at different pH values is shown in Figure 3. A considerable decrease in emission intensity was noticed as we moved from physiological pH to acidic pH values (pH 8.0–2.0). The decrease in emission intensity may have been due to the quenching mechanism, the protonation of water molecules, or acidic amino acids present in the surrounding intrinsic fluorophores. Similarly, a considerable decrease in emission intensity was noticed as we moved from physiological pH to basic pH values (pH 8.0–12.0), which might have been due to the deprotonation of basic amino acids that surrounded the intrinsic fluorophore and caused fluorescence quenching. The protonation and deprotonation changed the charge in the local milieu by altering electrostatic interactions and internal salt bridges that were present in native protein [43]. The emission spectra of the protein at pH 8.0 showed an emission maxima peak (λ_max_) at 344 nm, which is the native conformation of the protein. The plot of λ_max_ as a function of pH showed no considerable change in the λ_max_ of the protein from pH 4.0–9.0. This can be attributed to the microenvironment around the aromatic amino acid not being disturbed significantly. However, in highly acidic conditions (pH 2.0–3.0) and highly basic conditions (pH 10.0–12.0), a slight redshift of 2 nm from 344 to 346 in the λ_max_ was observed (inset of Figure 3). A redshift in emission maxima is indicative of the increased solvent interactions of aromatic amino acids due to the unfolding of the protein [44]. Previously, we reported that the wild-type G protein showed a redshift of 5 nm in the λ_max_ only at pH 12.0 [45]. Finally, our fluorescence study concluded that the tertiary structure of the protein was perturbed as we moved from the physiological pH to the acidic or basic pH values, and changes in the λ_max_ were observed in highly basic and acidic pH conditions.

#### 2.3.2. ANS Binding Measurements

Changes in the tertiary structure of a protein in various environmental conditions often lead to the exposure of hydrophobic patches, which are usually buried in the native form of the protein. Some proteins form an intermediate state during the process of unfolding, which is commonly known as the molten globule (MG) state [46]. ANS dye was specifically used to investigate the presence of hydrophobic clusters on the surface of the protein. ANS generally does not bind to a native protein, as hydrophobic patches are buried inside the core of the protein. ANS also does not bind to a denatured protein, as hydrophobic residues are present at large distances. The binding of ANS to the hydrophobic patches often leads to higher fluorescence intensity, which demonstrates the presence of an intermediate state, and hints at the formation of molten or premolten globule confirmations [46]. Figure 4 shows that the ANS fluorescence intensity of the protein at pH 2.0 and 3.0 was very high in comparison to the native state of the protein (pH 8.0). The high fluorescence intensity and shifting of emission maxima (λ_max_) toward the shorter wavelength suggested the exposure of hydrophobic patches to solvent [47]. Therefore, the non-native state of the mutant G protein at pH 2.0 and 3.0 was regarded as an acid-induced molten globule like state. Similar to the mutant protein, the wild–type G protein also formed the molten globule like state at highly acidic pH values in our previous investigation [45]; in both cases, this was attributed to the hydrophobic patches being exposed when the protein was populated at pH 2.0 and 3.0 in the solutions. It was observed that the ANS fluorescence intensity decreased as the pH of the solution moved toward the basic condition (inset of Figure 4). ANS binding was prevented in physiological and alkaline conditions due to the inaccessibility of the hydrophobic cluster, because at these pH values, the hydrophobic patches might have been buried in the core of the protein, or present at a large distance [48].

#### 2.3.3. GdmCl- and Urea-Induced Denaturation

The stability of a protein can be quantified by equilibrium unfolding measurements in the presence of urea or GdmCl [49,50]. The aromatic amino acid of the protein, which is often fully or partially buried in the hydrophobic core of the protein, serves as an indicator of the structural integrity of the protein. The stability of the protein was measured using intrinsic tryptophan, which is an important probe. Figure 5 and Figure 6 show the emission spectra of the mutant G protein with increasing concentrations of GdmCl and urea, respectively. We observed that the emission spectra of the protein changed as we increased the concentration of GdmCl and urea. As we increased the concentration of the denaturant, the fluorescence intensity decreased with the shifting of the emission maxima (λ_max_) toward the higher wavelength (redshift). The spectra of the protein in the absence of a denaturant showed an emission maxima (λ_max_) peak at 344 nm; however, at a higher concentration of denaturant, the λ_max_ of the protein was shifted to 356 nm. From these results, we inferred that the aromatic amino acids of the protein were shifted from nonpolar to polar environmental conditions, as the GdmCl and urea (denaturants) exposed the buried aromatic amino acid residues [51]. Our results also suggested that as we increased the GdmCl or urea concentration, unfolding of the protein occurred, which exposed the buried tryptophan residues to a more polar buffer condition. The alteration in the microenvironment of tryptophan was monitored using *F*_344_ (fluorescence emission intensity at 344 nm) as a function of GdmCl (inset of Figure 5) and urea (inset of Figure 6). The plots of *F*_344_ versus [GdmCl] and [urea] showed that the denaturation process in the presence of the denaturant followed a two-state process. The denaturation transition curves were examined to obtain the value of stability parameters such as ∆*G_D_*^0^ (Gibbs free energy change in the absence of denaturants), *m* (slope), and *C_m_* (transition midpoint of denaturation curve) by fitting the entire denaturation curve to Equation (1). The thermodynamic stability parameters of mutant G protein are given in Table 1.

In our previous investigation, we reported the GdmCl- and urea-induced denaturation of the wild-type G protein [45], in which we found that GdmCl- and urea-induced denaturation followed a two-state transition mechanism. The ∆*G_D_*^0^ value of the wild-type G protein in the presence of urea was 3.76 ± 0.34 kcal mol^−1^; however, it was found to be 2.87 ± 0.21 kcal mol^−1^ in the mutant G protein, which indicated that the wild-type G protein was more stable than the mutant G protein in the presence of urea. The *C_m_* value for the mutant G protein was found to be 2.90 ± 0.13 M, while it was 4.42 ± 0.16 M in the case of the wild-type G protein [45]. It is well known that urea interacts differently with hydrophobic groups than with either protein backbones or hydrophilic groups [52]. It should be noted that the unfolding process by urea arises due to the weakening of hydrophobic interactions between the polymer groups [53]. Urea denatures the protein by destabilizing the hydrophobic forces. It seemed that the mutant protein was less stable where hydrophobic forces were lower than those of the wild-type protein. On the other hand, the ∆*G_D_*^0^ value of the wild-type G protein in the presence of GdmCl was 2.53 ± 0.20 kcal mol^−1^; however, it was found to be 2.22 ± 0.22 kcal mol^−1^ in the mutant G protein, which we attributed to the wild-type G protein being more stable than the mutant G protein in the presence of GdmCl. The *C_m_* values of the wild-type (1.52 ± 0.07 M) and mutant protein (1.48 ± 0.08 M) were found to be similar in the presence of GdmCl; i.e., ~1.5 M [45]. The unfolding transitions induced by GdmCl and urea are not always found to be similar; the difference in the free energy of unfolding might have been due to the ionic effect of GdmCl [54].

### 2.4. Binding Studies of Heparan Sulfate with the Mutant G Protein 

#### 2.4.1. Fluorescence Binding Measurements

Fluorescence spectroscopy is a very simple and sensitive technique to study protein–ligand interactions. It is mainly used to determine the number of binding sites (*n*), Stern-Volmer constant (*K_sv_*), binding constant (*K*), and interaction mechanism between a protein and ligand. For binding measurements, the protein sample was excited at 280 nm. Excitation of a protein at 280 nm is considered as fluorescence of tyrosine, tryptophan, and phenylalanine [55]. Figure 7A shows the emission spectra of the mutant G protein with an increasing (0–40 µM) concentration of heparan sulfate (HS). The HS did not flourish alone, whereas the protein showed a peak of maxima (λ_max_) at 344 nm in the same environment. The fluorescence intensity of the protein gradually decreased as we increased the concentration of HS, suggesting the formation of a complex between the ligand and protein. The experimental data were analyzed to calculate the *K_sv_* using Equation (2). Figure 7B represents the Stern–Volmer plots of protein quenching in various concentrations of heparan sulfate. The *K_sv_* value was obtained using Equation (2) by fitting the fluorescence intensity ratio *F*_0_/*F* for various concentrations [C] of HS. The quenching mode was further confirmed from the value of *K_q_* (bimolecular quenching constant) using Equation (3). The experimental data were further analyzed by using Equation (4), which showed the binding constant (*K*) value (Figure 7C). The binding interaction parameters of HS with the mutant G protein are given in Table 2.

The binding constant (*K*) value of the mutant protein with HS was found to be 2.08 × 10^5^ M^−1^. However, in our previous study, we observed that the binding constant value between HS and the wild-type G protein was 3.98 × 10^6^ M^−1^ [45]. From these observations, we concluded that when we substituted the lysine residue with alanine (K117A) in the CX3C motif of the G protein, a low binding constant value was found. The low binding of the mutant G protein with HS was attributed to the mutation in this region disrupting the binding interactions. Moreover, the low binding might have been due to the disruption of electrostatic interactions between the positively charged amino acid (lysine) of the protein and the negatively charged group of the HS.

#### 2.4.2. Absorbance Binding Measurements

In a protein, the presence of a conjugated bond system in the side chain of aromatic amino acids acts as a chromophore that absorbs UV light in the range of 240–340 nm [56]. The mutant G protein contains five tyrosine residues, which showed an absorption maxima peak at 278 nm. The alteration in the tertiary structure of the protein with the addition of HS indicated binding of the ligands to the target protein [57]. The absorption spectra of mutant protein gradually decreased as we increased the concentration of HS, suggesting the binding of the ligand with the protein (Figure 8). It was noted that the quenching stopped after the addition of 48 µM of HS, which was an indication of the formation of a stable complex (inset of Figure 8). Similar to the mutant protein, the absorbance spectra of wild-type G protein decreased with an increasing concentration of HS, as reported in our previous investigation [45].

#### 2.4.3. ITC Measurements

For the binding affinity measurements of heparan sulfate with the mutant G protein, we performed isothermal titration calorimetry (ITC), which provides information about the interaction of the ligand with the protein. Figure 9 shows the thermogram of raw data in power versus time for titration of the ligand (HS) against the reaction cell that contained the mutant G protein. In the upper panel, each peak in the binding isotherm signifies the single injection of the HS, whereas the lower panel denotes the integration of the area under each injection peak of the heat profile, which aided in generating a differential curve. The thermodynamic binding parameters of HS with the mutant G protein are given in Table 3.

The heat profile of the mutant G protein was exothermic with a negative heat pulse, indicating the protein–HS binding pattern. The changes in enthalpy (∆*H*^0^) and the Gibbs free energy (∆*G*^0^) were largely negative, which suggested the spontaneous nature of the reaction. The negative values of ∆*G*^0^ and ∆*H*^0^ showed that the binding of HS with the protein was mainly driven by the electrostatic interaction. The binding affinity (*K_a_*) value of HS with the mutant G protein was found to be 2.51 × 10^4^ M^−1^. However, in our previous investigation, we found a higher binding affinity with the wild-type G protein; i.e., 10.7 × 10^4^ M^−1^ [58]. From these observations, we concluded that after the substitution of lysine (K) with alanine (A) in the CX3C motif of the G protein, a low binding value was observed. The low binding of the mutant G protein with HS suggested that mutation in this region altered the binding interactions. This observation was consistent with our fluorescence quenching results, in which we also observed low binding. A previous study demonstrated that RSV bound to the corresponding chemokine receptor (CX3CR1) via the CX3C chemokine motif (^182^CWAIC^186^) of the G protein, which contributes to disease pathogenesis [15]. A recent study also showed that mutation in the CX3C motif by insertion of alanine (A^186^), mutating it to CX4C (^182^CWAIAC^187^), which is known to block binding to the CX3CR1, might decrease disease and immune modulation associated with the G protein of RSV [37]. The results obtained from the ITC may help to explain the binding mechanism of the virion with the host cell.

We observed a significant difference between the thermodynamic parameters of fluorescence and ITC. It is well known that fluorescence measures only the local changes around the microenvironment of the fluorophore upon ligand binding, whereas on the other hand, ITC monitors the global changes in terms of heat released or absorbed during the breaking or formation of bonds upon ligand–protein interactions [59]. Variations in thermodynamic parameters derived from fluorescence and ITC have been seen in several other ligand–protein interaction studies [60,61,62].

#### 2.4.4. Molecular Docking

Molecular docking of heparan sulfate with the mutant G protein was performed to determine the binding energy, as well as to identify the specific amino acid residues involved in ligand binding and their intramolecular distances. The heparan sulfate occupied the binding site pocket of the mutant G protein and formed a single hydrogen bond with Asn1127, Pro132, Thr133, Ser191, and Glu226, and triple hydrogen bonds with Thr134. In addition, HS also formed two carbon–hydrogen bonds with Pro132 and Arg196 (Figure 10A,C). The intramolecular distance between the ligand and protein was found to be in the range of 2.03 Å to 2.77 Å (Figure 10B). The HS showed a binding energy of −6.4 kcal/mol with the mutant G protein. The binding constant (*K_b_*) value was calculated using the value of the binding energy, and was found to be 4.9 × 10^4^/mol. The binding parameters of HS with the mutant G protein are given in Table 4.

However, in our previous investigation, the wild-type G-protein binding energy was −6.8 kcal/mol, and the key interacting residues Cys116, Lys117, and Arg196 were mainly involved in a binding interaction [45]. From these observations, we found that when we substituted the lysine residue with alanine (K117A) in the CX3C motif of the G protein, a low binding energy was found. The low binding of the mutant (K117A) G protein with HS was attributed to that mutation in this region disrupting the binding interactions. It was interesting to note that when we mutated the lysine to alanine, it did not form any hydrogen bond with Ala117. However, in the wild-type G protein, HS directly formed a hydrogen bond with Lys117. From the binding study, we also found that the mutated G protein did not form any carbon–hydrogen bond with Leu115 and Cys116, which were present in the wild-type G protein. From these observations, we concluded that besides Lys117, Leu115 and Cys116 might also play an important role in the binding of the wild-type G protein that was absent in the mutant (K117A) protein.

Since we used only the ectodomain part of the G protein for our study, the mutated amino acid position in the CX3C motif was K117A. However, the mutated amino acid position with respect to the full-length G protein was K183A. Previously, Verga and colleagues created a synthetic peptide with alanine mutation at all positions from Ile183 to Lys195, and their study showed that Ile185 and Arg188 were very important in the recognition of lung mononuclear cells from BALB/c mice that were immunized by the RSV G protein [63]. In another study, Huang and colleagues showed that amino acids Arg188 and Lys192 were very important in providing protective immunity against RSV, as well as in induction of RSV-associated eosinophilia in BALB/c mice [64].

The binding of HS with the mutant protein determined by ITC and fluorescence measurements further complemented our molecular docking results. All the binding studies suggested that the substitution of lysine with alanine in the CX3C motif of G protein effectively inhibited the interaction with HS. The binding parameters of the mutant G protein with HS obtained from fluorescence quenching, ITC, and molecular docking are given in Table 4.

Here, we propose that instead of the wild type, the mutant G protein may be used for vaccine development, because the mutant protein may not bind to HS effectively. The binding parameters of the wild-type and mutant G proteins with HS are given in Table 5. It must be noted that there was no shift in the fluorescence intensity of the mutant and wild-type proteins at physiological pH. Hence, we assumed that the mutation in the CX3C motif of G protein reduced the binding without affecting its tertiary structure.

## 3. Materials and Methods

### 3.1. Materials

Sodium chloride, imidazole, GdmCl, urea, glycine, ethanol, Tris buffer, etc. were brought from Merck (Darmstadt, Germany). Glycerol, *N*-lauroylsarcosine, and Triton X-100 were purchased from Sigma (St. Louis, MI, USA). Luria–Bertani broth, kanamycin, and imidazole were brought from Himedia, India. All the chemicals used for the experiments were of analytical grade.

### 3.2. Cloning, Expression and Purification of Mutant G Protein

The full-length G-protein gene sequence (accession no. KJ690590) was taken from the NCBI database. For our study, only the ectodomain part of the G protein was taken. The ectodomain region of the G-protein gene with a mutation in the CX3C motif (K117A) was codon-optimized and inserted in the pUC57 vector by commercial services (Gene Script, Piscataway, NJ, USA). The gene was further subcloned in the pET28a expression vector. The recombinant expression vector was transformed into the *E. coli* (DH5α) strain, and the plasmid was isolated using a commercially available plasmid isolation kit and analyzed through agarose gel electrophoresis. The plasmid was further transformed and expressed into the *E. coli* strain of BL23 (DE3), and the protein was purified using Ni-NTA affinity chromatography with some changes described previously [58,65,66]. Briefly, the protein was expressed at 30 °C for 12 h with induction of 0.5 mM IPTG. The inclusion bodies (IBs) were prepared using a standard protocol [58] and solubilized in buffer (50 mM CAPS buffer pH 11.0, 100 mM NaCl, and 0.5% *N*-lauroylsarcosine), and the protein was purified by Ni-NTA affinity chromatography. The desired protein fraction was eluted with an increasing concentration of imidazole. The fractions of eluted protein were analyzed through sodium dodecyl sulphate polyacrylamide gel electrophoresis (SDS-PAGE). The desired eluted sample was dialyzed against NaCl (100 mM) and Tris buffer (20 mM) at pH 7.5. The buffer was repeatedly changed at least 5 times in 24 h at 4 °C to obtain the refolded protein. The concentration of protein was measured by a molar absorbance coefficient (ε) of 8730 M^−1^ cm^−1^ at 280 nm using Jasco V-600 UV–visible spectrophotometer [67].

### 3.3. Sample Preparation

A wide range of buffers was used to determine the pH-dependent alteration in the structure of the protein. For pH 2.0 and 3.0, glycine–HCl was used; and for pH 10.0, 11.0, and 12.0, glycine–NaOH buffer was prepared. For pH 4.0 and 5.0, acetate buffer was prepared. For pH 6.0 and 7.0, phosphate buffer was prepared. For pH 8.0 and 9.0, Tris buffer was prepared. Before performing the spectral measurements, the sample was incubated for at least 2–3 h to attain equilibrium. A stock solution of GdmCl (8.7 M) and urea (10.5 M) was prepared to observe the structural stability of the protein. The stock solution of urea and GdmCl was prepared in 25 mM Tris buffer at pH 8.0. The calculated amount of buffer, protein, and denaturant (urea/GdmCl) was mixed and incubated for 3–4 h at room temperature to ensure the completion of the denaturation process. All the experiments were performed in triplicate, and the blank values were subtracted from each measurement of the samples.

### 3.4. Fluorescence Measurements

The fluorescence emission spectra of the mutant protein were recorded in a Jasco spectrofluorometer (FP6200) with a quartz cuvette of 1 cm path length at 25 ± 1 °C. We observed alterations in the emission spectra of the mutant G protein in a wide range of buffers (pH 2.0–12.0) and different concentrations of urea and GdmCl. For spectral measurements, the entrance and exit slit widths were set at 5 nm and 10 nm, respectively. The protein sample was excited at 280 nm, and we collected the emission spectra in the wavelength region of 300–400 nm.

### 3.5. ANS Measurements

The ANS fluorescence was performed with a Jasco spectrofluorometer (FP6200) having a quartz cuvette of 1.0 cm path length at 25 ± 1 °C. For ANS fluorescence measurements, the protein-to-ANS ratio was taken at 1:20. For spectral measurements, the excitation and emission slit widths were set at 5 nm. The ANS sample was excited at 380 nm, and we collected the spectra in the wavelength range of 400–600 nm. Before performing the spectral measurements, the prepared samples were incubated at least for 30 min in the dark.

### 3.6. Denaturation Spectral Measurements

The transition curves generated after plotting the spectral property *y* (*F*_344_) against the molar concentration of GdmCl/urea were evaluated for the determination of protein stability. Thermodynamic properties such as *m* and *C_m_* were used to calculate the stability of the protein, where Δ*G_D_*^0^ is the Gibbs free energy change in the absence of denaturants, *m* is the slope (*∂*Δ*G_D_*/*∂*[*GdmCl*/*urea*], and *C_m_* (= Δ*G_D_*^0^/*m*) is the transition midpoint of the denaturation curve where ∆*G_D_* = 0. Least-square approaches were used to fit the denaturation transition curve by the following equation:(1)$$y=\frac{y_{N}+y_{D}\times Exp\left[-\left(\Delta G_{D}{}^{0}-m\left[urea/GdmCl\right]\right)/RT\right]}{\left(1+Exp\left[-(\Delta G_{D}{}^{0}-m\left[urea/GdmCl\right])/RT\right]\right)}$$
where *y_N_* and *y_D_* represent the anticipated optical properties of the native and denatured protein, respectively, under the same experimental condition in which *y* was recorded; *R* represents the universal gas constant; and *T* represents the temperature in Kelvin.

### 3.7. Fluorescence Binding Measurements

The fluorescence binding studies of heparan sulfate with the mutant G protein were performed by a Jasco spectrofluorometer (FP6200) with a quartz cuvette of 1 cm path length at 25 ± 1 °C. The stock solution of HS was prepared in 20 mM Tris buffer (pH 7.5). An increasing concentration of HS (2–40 µM) was used to titrate against the constant concentration of the protein. The protein sample was excited at 280 nm, and we collected the spectra at 300–400 nm. For spectral measurements, the entrance and exit slit widths were set at 5 nm and 10 nm, respectively. The blank spectra (buffer and HS) were subtracted from the titrated samples to obtain the final spectra of the protein.

The fluorescence binding of HS with mutant G protein was examined to determine the values of the number of binding sites (*n*), Stern–Volmer constant (*K_sv_*), and binding constant (*K*).

Equation (2) was used to find the Stern–Volmer constant by analyzing the quenching data:(2)$$\frac{F_{0}}{F}=1+K_{SV}\left[C\right]$$
where *F*_0_ represents the protein intensity in the absence of HS, *F* represents the protein intensity at a particular concentration of heparan sulfate at 344 nm, [*C*] represents the various concentrations of HS, and *K_SV_* represents the Stern–Volmer binding constant.

Equation (3) was used to determine the bimolecular quenching constant (*K_q_*) to check the mode of quenching in the protein–ligand complex:(3)$$K_{q}=\frac{K_{SV}}{\tau_{o}}$$
where *τ*_0_ is the average integral fluorescence lifetime of tryptophan (2.7 × 10^−9^ s).

Further, Equation (4) was used to determine the modified Stern–Volmer constant (double log plot) that gave the value of the binding constant (*K*) in the protein–ligand complex:(4)$$log\left(\frac{F_{0}-F}{F}\right)=logK+n$$
where *n* represents the number of binding sites, and *K* represents the binding constant of the protein–ligand complex.

### 3.8. Absorbance Binding Measurements

The spectral measurement of heparan sulfate with the mutant G protein was performed with a Jasco UV/visible spectrophotometer (V-660) using 1 cm path length cuvettes. A constant concentration of protein was titrated with an increasing concentration of heparan sulfate (2–56 µM), and we collected the spectra in the wavelength range of 240–340 nm.

### 3.9. ITC Measurements

The binding studies of the heparan sulfate with the mutant G protein were also performed by isothermal titration calorimetry at 25 °C using VP-ITC (MicroCal, Northampton, MA, USA). The sample cell was injected at a 1:20 ratio with 10 µM protein and 200 µM heparan sulfate for titration. The 10 µL aliquots of HS were loaded in every step at an interval of 300 s from the syringe. The stoichiometry value (*n*), binding constant (*K_a_*), enthalpy change (∆*H*^0^), and entropy change (∆*S*^0^) were determined from the measured heat change (δ*Hi*) upon the interaction of the protein with heparan sulfate. The obtained raw data were accessed and analyzed using MicroCal Origin 8.0 software (MicroCal, Northampton, MA, USA). Using Equation (5), we calculated the Gibbs free energy change (∆*G*^0^) using the thermodynamic parameters obtained above:(5)$$\Delta G^{0}=-RT\;\ln K_{a}=\Delta H-T\Delta S$$

### 3.10. Molecular Docking

The docking studies were performed to determine the interaction of heparan sulfate with the mutant G protein. The crystal structure of the G protein has yet to be determined. Hence, for the docking study, we modeled the three-dimensional structure of the ectodomain G protein using in silico methods. In our earlier studies, we described the modeled structure of the protein in detail [68]. The point mutation was generated in the CX3C motif of the ectodomain G protein at position 117 by substituting the lysine residue with alanine (K117A) using PyMOL software. Heparan sulfate’s chemical structure was retrieved from the PubChem database and converted into a PDBQT file using the Open Babel application of PyRx. Docking and visualization were carried out using bioinformatics tools such as PyRx (accessed on 15 December 2021), PyMOL (Schrödinger, Inc., New York, NY, USA, 2010) and Discovery studio software (Dassault Systèmes BIOVIA, San Diego, CA, USA, 2017) [69,70]. The docking was performed by forming the grid box in such a way as to occupy the active binding site of the protein. The docking study was performed structurally blind, which meant that the molecule was free to move around and search the binding sites of the protein with the most favorable and minimum energy conformation. Based on the binding energy, the best docked structure was taken and analyzed using Discovery studio software. The binding constant (*K_b_*) value for the ligand–protein interaction was calculated using Equation (6):(6)$$\Delta G=-RT\;\ln K_{b}$$
where *R* is the gas constant (1.98719 cal/mol), *T* is the temperature in Kelvin (298.15 K), and ∆*G* is the docking energy.

## 4. Conclusions

In conclusion, this study demonstrated that a mutation in the CX3C motif of the G protein disrupted the binding of heparan sulfate with the mutant protein as compared to the wild-type G protein. The obtained experimental results demonstrated that targeting the mutation in the G protein may be an effective strategy to counteract the inhibition of host–pathogen interaction. For the development of an effective and safer vaccine, this type of amino acid mutagenesis may be exploited to improve vaccine efficacy. The mutation in the CX3C motif of the G protein may be used as a platform to develop an effective and safer vaccine against RSV.

## Figures and Tables

**Figure 1 ijms-23-01950-f001:**
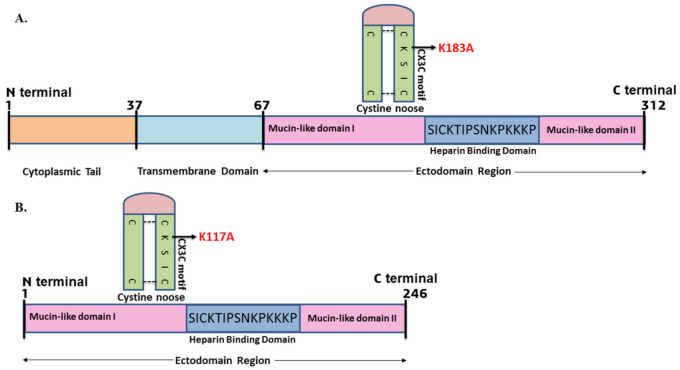
(**A**) Schematic representation of RSV G protein showing mutation in the CX3C motif of G protein (K183A) with respect to full length. (**B**) Considering the ectodomain region of the G protein, the mutated amino acid position was K117A.

**Figure 2 ijms-23-01950-f002:**
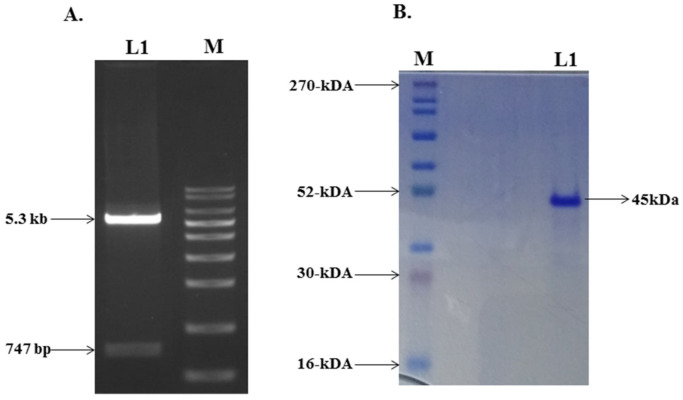
(**A**) Lane 1 showing the digested products of pET–28a vector and insert of mutant G protein gene. Lane M is a molecular marker. (**B**) SDS–PAGE gel photograph of purified mutant G protein.

**Figure 3 ijms-23-01950-f003:**
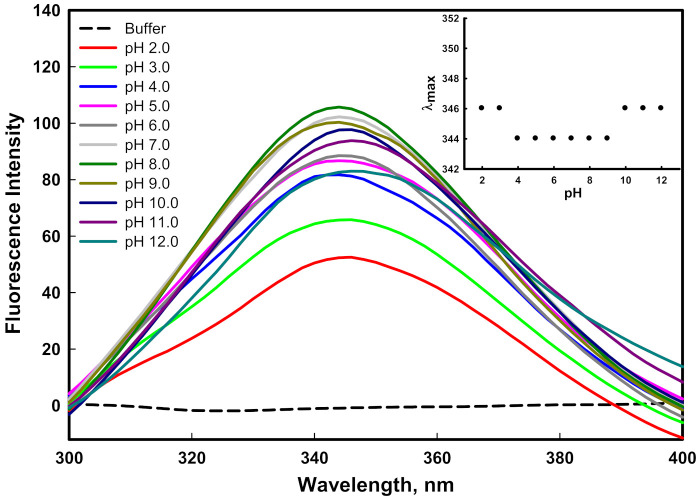
Fluorescence spectra of mutant G protein at different pH values ranging from 2.0–12.0 at 25 °C. The inset illustrates the protein denaturation profile by monitoring changes in emission maxima (λ_max_) as a function of pH.

**Figure 4 ijms-23-01950-f004:**
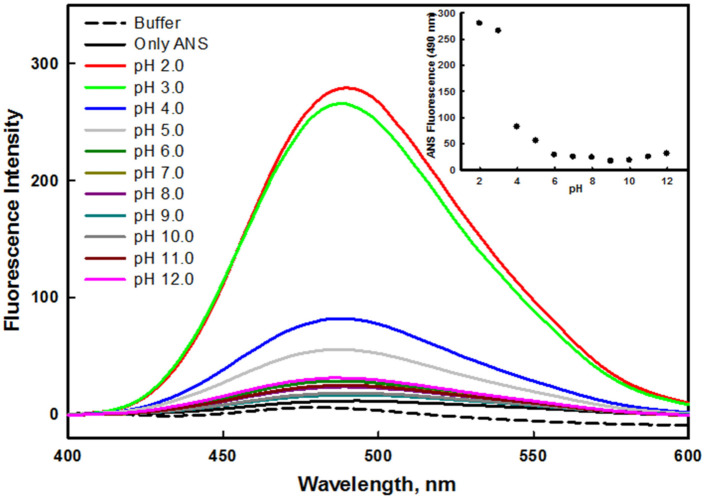
ANS fluorescence spectra of mutant G protein at different pH values ranging from 2.0–12.0 at 25 °C. The inset illustrates the ANS profile of protein by observing changes in *F*_490_ as a function of pH.

**Figure 5 ijms-23-01950-f005:**
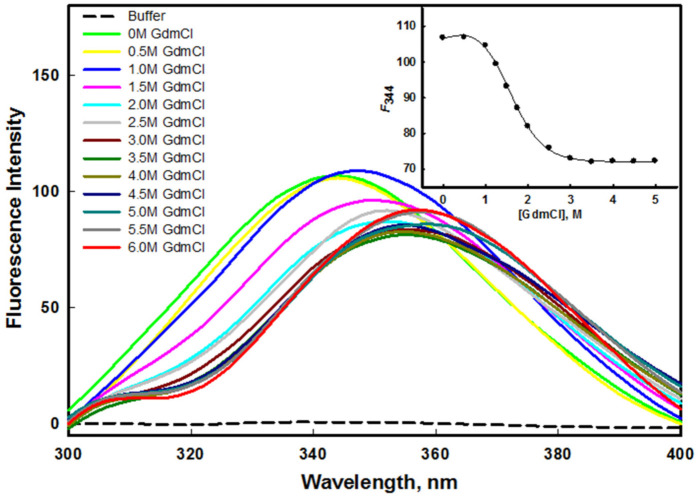
GdmCl–induced denaturation of mutant G protein at pH 8.0 and 25 °C. The inset shows the denaturation curve of protein (plot of *F*_344_ as a function of different concentration of GdmCl).

**Figure 6 ijms-23-01950-f006:**
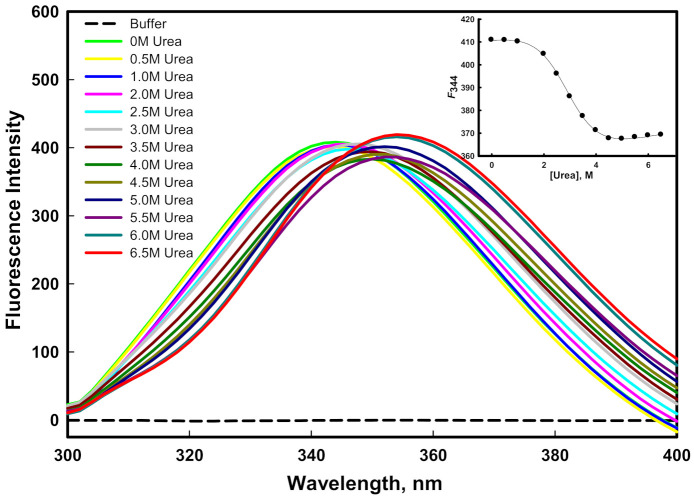
Urea-induced denaturation of mutant G protein at pH 8.0 and 25 °C. The inset shows the denaturation curve of protein (plot of *F*_344_ as a function of different concentration of urea).

**Figure 7 ijms-23-01950-f007:**
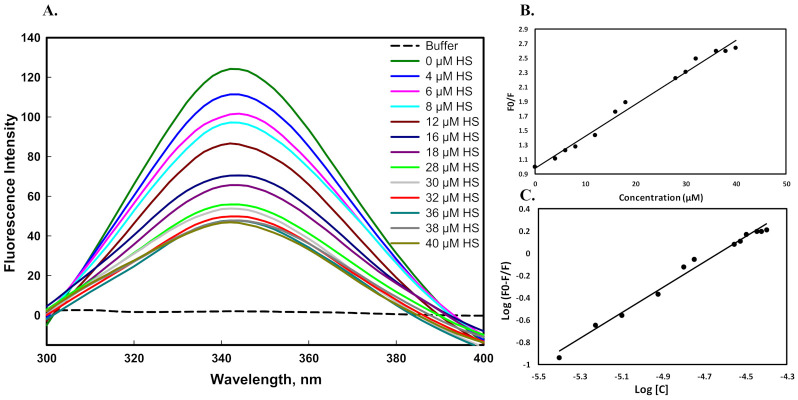
Fluorescence binding studies of the mutant G protein with heparan sulfate at pH 7.5 and 25 °C. (**A**) Fluorescence spectra of mutant G protein with increasing concentration of heparan sulfate (0–40 µM). (**B**) Stern–Volmer plot for quenching of protein–HS complex. (**C**) Modified Stern–Volmer plot obtained from titration of heparan sulfate, which is used for the calculation of binding affinity.

**Figure 8 ijms-23-01950-f008:**
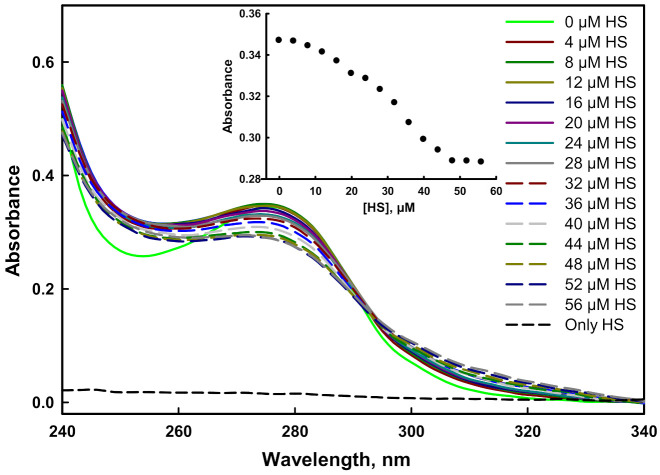
Absorbance binding measurements of the mutant G protein with heparan sulfate at pH 7.5 and 25 °C. Spectra were recorded with increasing concentrations of heparan sulfate (0–56 µM).

**Figure 9 ijms-23-01950-f009:**
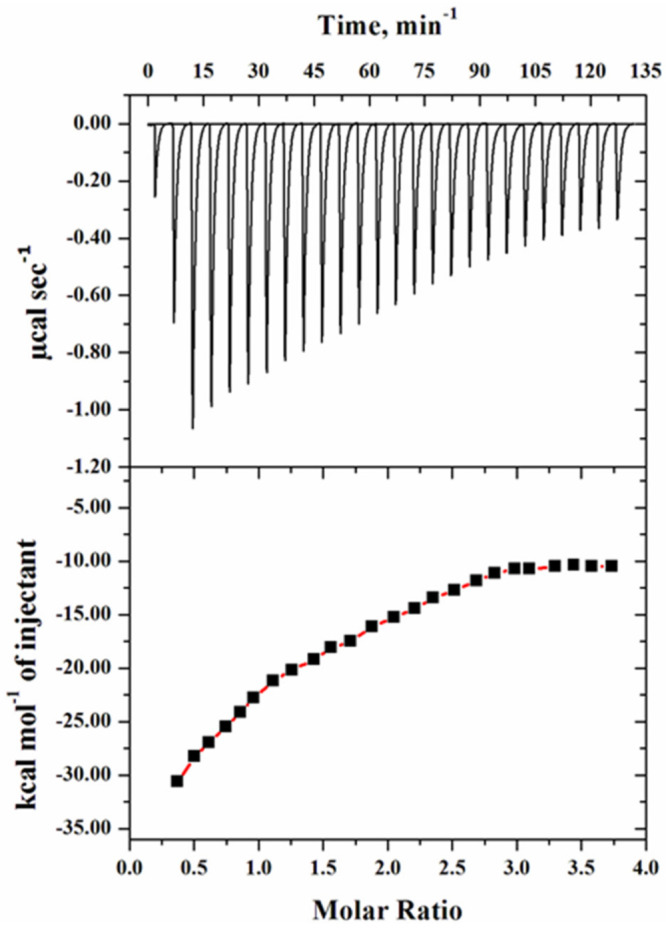
ITC thermogram of mutant G protein with 10 µM protein and 200 µM heparan sulfate. The thermogram of raw data in power versus time is shown in the upper panel, with each peak in the binding isotherm representing a single ligand injection. The lower panel indicates the quantity of heat emitted as a function of the mole ratio of the protein to ligand at pH 7.5 and 25 °C.

**Figure 10 ijms-23-01950-f010:**
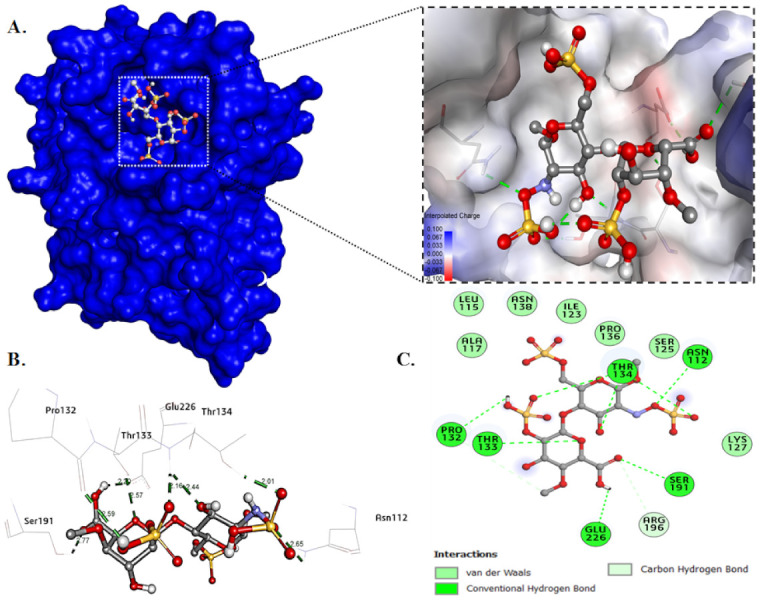
Molecular docking of mutant G protein with heparan sulfate. (**A**) Surface view of docked protein-HS complex. (**B**) Illustration of the protein-HS complex showing hydrogen bond distances. (**C**) Detailed two-dimensional plot illustrating the type of interactions between protein and ligand.

**Table 1 ijms-23-01950-t001:** Thermodynamic parameters obtained from GdmCl- and urea-induced denaturation of mutant G protein at pH 8.0.

Probes	Denaturants	Transition	∆*G_D_*^0^, kcal mol^−1^	*m*, kcal mol^−1^ M^−1^	*C_m_*, M
*F* _344_	GdmCl	N↔D	2.22 ± 0.22	1.50 ± 0.10	1.48 ± 0.08
	Urea	N↔D	2.87 ± 0.21	0.99 ± 0.06	2.90 ± 0.13

**Table 2 ijms-23-01950-t002:** Fluorescence binding parameters of the mutant G protein with heparan sulfate at pH 7.5 and 25 °C.

*K_sv_* (M^−1^)	*K_q_* (M^−1^ s^−1^)	*K* (M^−1^)	*n*	R^2^
4.42 × 10^4^	1.63 × 10^13^	2.08 × 10^5^	1.14	0.98

**Table 3 ijms-23-01950-t003:** Binding parameters of the mutant G protein with heparan sulfate by ITC measurements at pH 7.5 and 25 °C The data in the bracket is standard error of the given values.

Thermodynamic Binding Parameters (Units)				
*n*	*K_a_* (M^−1^)	∆*H*^0^ (cal mol^−1^)	∆*S*^0^ (cal mol^−1^ deg^−1^)	∆*G*^0^ (cal mol^−1^)
2.05 (± 0.49)	2.51 × 10^4^ (± 6.9 × 10^3^)	−9.08 × 10^4^ (± 3.17 × 10^3^)	−285	−5.95 × 10^3^

**Table 4 ijms-23-01950-t004:** Binding parameters of heparan sulfate with mutant G protein obtained from fluorescence, ITC, and molecular docking.

Compound	Fluorescence	ITC		Docking	
	Binding Constant (*K*) M^−1^	Binding Constant (*K_a_*) M^−1^	Binding Energy (∆*G*^0^) cal mol^−1^	Binding Constant *K_b_* (/mol)	Binding Energy (∆*G*) kcal/mol
Heparan sulfate	2.08 × 10^5^	2.51 × 10^4^	−5.95 × 10^3^	4.9 × 10^4^	−6.4

**Table 5 ijms-23-01950-t005:** Comparative binding parameters of heparan sulfate with wild-type and mutant G proteins obtained from fluorescence, ITC, and molecular docking.

Compound	Proteins	Binding Constant (*K*) M^−1^ (Fluorescence)	Binding Constant (*K_a_*) M^−1^ (ITC)	Binding Energy (∆*G*) kcal/mol (Docking)
Heparan sulfate	Wild type G protein	3.98 × 10^6^ [45]	10.7 × 10^4^ [58]	−6.8 [45]
	Mutant G protein	2.08 × 10^5^	2.51 × 10^4^	−6.4

## Data Availability

Not applicable.

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
