# Peer review of "Mutation in the CX3C Motif of G Protein Disrupts Its Interaction with Heparan Sulfate: A Calorimetric, Spectroscopic, and Molecular Docking Study"

_ijms, 2022, doi:10.3390/ijms23041950_

Round 1

Reviewer 1 Report

In the manuscript by Hamze et al. the authors study the structure, stability, and binding to heparan sulfate of point-mutant ectodomain of the respiratory syncytial virus G-protein.

Overall, the paper is well written. The results and discussions seem good.

In Figure 2 it would be quite informative if the authors could include the results of SDS-Page for wild-type G protein (or ectodomain)

Please explain why you decided to use two denaturing agents (urea and GdmCl) for the protein stability measurements? Did the authors expect differences in the two means used and why? Please elaborate reasons.

Figures 7 and 8: it is not clear which color indicates a certain concentration of heparan sulfate. A legend should be added to the figures.

There is great concern about the number of experiments performed - with the exception of fluorescence measurements, which the authors state were performed in triplicate, all other experiments appear to be performed in a single run. Authors should either explain the reasons or present experiments in iterations.

The authors have repeatedly argued that this mutated ectodomain of G-protein can be used to improve vaccine efficacy and safety, but no background or explanation of this statement is provided.

References: some references are cited irregularly, sometimes the journal is not cited, and some citations cannot be found (eg, 55).

Author Response

Authors’ response to reviewers’ comments

We are grateful to the esteemed reviewers for their critical evaluation of the manuscript. Their comments have indeed uplifted the quality of the manuscript. We are hereby providing a point-to-point response to the reviewer’s comments in addition to the revision of the manuscript as per the suggestion of the reviewers.

Response to Reviewer 1 Comments

Reviewer 1:

Comments and Suggestions for Authors

In the manuscript by Hamza et al. the authors study the structure, stability, and binding to heparan sulfate of point-mutant ectodomain of the respiratory syncytial virus G-protein.

Overall, the paper is well written. The results and discussions seem good.

Point 1: In Figure 2 it would be quite informative if the authors could include the results of SDS-Page for wild-type G protein (or ectodomain)

Response 1: It was a good suggestion from the Reviewer. The SDS-PAGE for wild type ectodomain G protein was reported earlier in our previous published article (Hamza A et al., ACS Omega, 2021). Moreover, we observed that wild type and mutant protein has similar size of band on SDS-PAGE analysis.

Point 2: Please explain why you decided to use two denaturing agents (urea and GdmCl) for the protein stability measurements? Did the authors expect differences in the two means used and why? Please elaborate reasons.

Response 2: Urea and GdmCl are the most frequently used protein denaturants. The main advantage of these denaturants is that the extent of unfolding, which is generally greater than can be achieved by other mean of denaturation. Urea denatures the proteins through favorable preferential interactions with protein side chains and backbone. However, GdmCl denatures proteins by inhibiting salt-bridge formations between charged amino acids. The unfolding transitions induced by GdmCl and urea are not always similar; the difference in the free energy of unfolding might be due to the ionic effect of GdmCl. In our study, we observed the Cm value in the presence of GdmCl was 1.48 M. However, the Cm value in the presence of urea was 2.90 M. Hence, we found GdmCl to be more potent denaturant as compared to the urea.

Point 3: Figures 7 and 8: it is not clear which color indicates a certain concentration of heparan sulfate. A legend should be added to the figures.

Response 3: The legends of figure 7 and 8 were added as per the suggestions in the revised version of the manuscript. We have also revised the legends of the figure 3 and 5.

Point 4: There is great concern about the number of experiments performed - with the exception of fluorescence measurements, which the authors state was performed in triplicate, all other experiments appear to be performed in a single run. Authors should either explain the reasons or present experiments in iterations.

Response 4: The structural characterization of protein in wide range of pH and stability measurements in the presence of urea and GdmCl was performed in triplicate. The needful changes have been incorporated in the revised version of the manuscript.

Point 5: The authors have repeatedly argued that this mutated ectodomain of G-protein can be used to improve vaccine efficacy and safety, but no background or explanation of this statement is provided.

Response 5: The background statement for vaccine efficacy and safety has been incorporated in the introduction section of the revised manuscript.

Point 6: References: some references are cited irregularly, sometimes the journal is not cited, and some citations cannot be found (eg, 55).

Response 6: As per suggestion of reviewer, references were corrected wherever needed and incorporated in the revised version of the manuscript.

Reviewer 2 Report

The article presents an interesting study that integrates experimental and molecular simulations studies aimed to characterize a mutated form of the ectodomain region of G protein from respiratory syncytial virus. The K183A substitution resulted able to maintain structural properties of the wt protein. At the same time, the mutation seems potentially able to reduce the binding to heparane, a feature related to the infection mechanism of the virus. The authors propose to  use the mutated protein for vaccine development.
The study is interesting and well designed, suitable for publication after the assessment of some minor points.
- A more clear comparison to the properties already determined for the wild-type protein is needed, also by graphical visualization of the properties of wt in comparison to what is shown in figure 3-9 and in tables.
- Table 4 and 5 can be modified in an unique table.

Author Response

Authors’ response to reviewers’ comments

We are grateful to the esteemed reviewers for their critical evaluation of the manuscript. Their comments have indeed uplifted the quality of the manuscript. We are hereby providing a point-to-point response to the reviewer’s comments in addition to the revision of the manuscript as per the suggestion of the reviewers.

Response to Reviewer 2 Comments

Reviewer 2:

Comments and Suggestions for Authors

The article presents an interesting study that integrates experimental and molecular simulations studies aimed to characterize a mutated form of the ectodomain region of G protein from respiratory syncytial virus. The K183A substitution resulted able to maintain structural properties of the wt protein. At the same time, the mutation seems potentially able to reduce the binding to heparan, a feature related to the infection mechanism of the virus. The authors propose to use the mutated protein for vaccine development. The study is interesting and well designed, suitable for publication after the assessment of some minor points.

Point 1: A more clear comparison to the properties already determined for the wild-type protein is needed, also by graphical visualization of the properties of wt in comparison to what is shown in figure 3-9 and in tables.

Response 1:  We have incorporated the data in the tabular form (Table 5) for the comparison between wild type and mutant protein in the revised manuscript as per suggestion of Reviewers. The graphical visualization of both wild type and mutant protein is given in the form of graphical abstract. Moreover, the comparative studies between the wild type and mutant has been mentioned in every section of this study with the reference of data (wild type) published earlier (Hamza A et al., Molecules, 2021 and ACS Omega, 2021).

Point 2: Table 4 and 5 can be modified in an unique table.

Response 2: As per the suggestion of the reviewer, the table 4 and 5 are modified and incorporated in the revised version of the manuscript.